# Chemotherapy-Induced Survivin Regulation in Acute Myeloid Leukemia Cells

**Petra Otevřelová and Barbora Brodská ***

Department of Proteomics, Institute of Hematology and Blood Transfusion, U Nemocnice 1, 12820 Prague, Czech Republic; otevrelova@uhkt.cz
* Correspondence: brodska@uhkt.cz

**Abstract:** Survivin is a 16.5 kDa protein highly expressed in centrosomes, where it controls proper sister chromatid separation. In addition to its function in mitosis, survivin is also involved in apoptosis. Overexpression of survivin in many cancer types makes it a suitable target for cancer therapy. Western blotting and confocal microscopy were used to characterize the effect of chemotherapy on acute myeloid leukemia (AML) cells. We found enhanced survivin expression in a panel of AML cell lines treated with cytarabine (Ara-C), which is part of a first-line induction regimen for AML therapy. Simultaneously, Ara-C caused growth arrest and depletion of the mitotic cell fraction. Subsequently, the effect of a second component of standard therapy protocol, idarubicin, and of a known survivin inhibitor, YM-155, on cell viability and survivin expression and localization in AML cells was investigated. Idarubicin reversed Ara-C-induced survivin upregulation in the majority of AML cell lines. YM-155 caused survivin deregulation together with a viability decrease in cells resistant to idarubicin treatment, suggesting that YM-155 might be efficient in a specific subset of AML patients. Expression levels of other apoptosis-related proteins, in particular X-linked inhibitor of apoptosis (XIAP), Mcl-1, and p53, and of the cell-cycle inhibitor p21 considerably changed in almost all cases, confirming the off-target effects of YM-155.

**Keywords:** survivin; cytarabine; idarubicin; YM-155; acute myeloid leukemia

## 1. Introduction

Survivin (baculoviral inhibitor of apoptosis repeat-containing 5, BIRC5) is the smallest member of the inhibitor of apoptosis (IAP) protein family [1]. It is coded by the BIRC5 gene and its molecular weight is 16.5 kDa. Survivin is overexpressed in many cancers, e.g., head-and-neck squamous cell carcinoma [2], ovarian [3] or gastric tumors [4], and hematologic malignancies [5], and its overexpression is associated with poor prognosis. On the other hand, survivin is almost undetectable in differentiated healthy cells, making it a promising target for cancer therapy [6].

Survivin has at least two independent functions [7]. It controls centrosome–microtubule assesment and microtubule-mediated sister chromatid separation during mitosis, and it plays an important role in apoptosis, in which it mediates inhibition of executive caspase 3/7 activity [8].

The two distinct functions of survivin determine its variable localization. In mitotic cells, survivin forms complexes with other chromosome passenger complex-related proteins, Borealin and Aurora B, and it localizes at kinetochores in metaphase, at midzone in anaphase, and at the chromatin bridge in telophase. The C-terminal part of the survivin molecule was found to be responsible for the proper localization of survivin during mitosis [9]. Conversely, the N-terminus is required for apoptosis induced by irradiation [9] and relocalization of survivin from the cytoplasm to the nuclei was observed during apoptosis [10].

The survivin promoter is directly regulated by tumor suppressor p53, and survivin overexpression is frequently detected in p53-deficient cells [11,12]. Drug-induced stabiliza-

tion of p53 protein often leads to survivin deregulation [8,13–15]. Interestingly, p53/nuclear factor (NF)-κB crosstalk during replication stress enhances survivin level and chemoresistance [16]. Survivin silencing induced p21 and p53 expression in breast carcinoma cells [17]. p21-mediated survivin suppression at the transcription level was found in HepG2 cells, likely through the block at gap 1 (G1)/G0 phase and downregulation of transcription factors E2F-1 and p300 [18]. Conversely, survivin binds Cdk4 from complex with p21, which releases the p21 for an interaction with procaspase 3 and blocks caspase-3 activity [19]. However, p21-induced survivin attenuation was observed only in cells arrested in G1/G0 phase, while enhanced survivin level was detected when p21 overexpression occurred during synthesis (S)–G2 transition [20]. Survivin expression is also regulated by transcription factors STAT3 and 5 [21,22] or β-catenin [12,23], which are important downstream molecules of fms-like tyrosine kinase 3 (Flt3) signalization [24].

Survivin interacts also with other members of the IAP family, particularly with X-linked inhibitor of apoptosis (XIAP). Cytoplasmic survivin stabilizes XIAP and enhances its antiapoptotic function [25]. Recently, it was reported that survivin overexpression in leukemia stem cells (LSCs), together with XIAP overexpression in granulocytes, causes a resistance of chronic myeloid leukemia (CML) cells to treatment with imatinib [26]. Survivin interaction with XIAP results in lower activity of caspase 9 [8]. On the other hand, survivin degradation in the proteasome is regulated by XIAP-mediated ubiquitinylation [16].

Survivin also contributes to DNA damage repair [17,27]. Moreover, it was reported to affect cell adhesivity [28] or intercellular communication [29].

Several studies reported on the positive correlation of high survivin expression with worse survival and relapse-free prognosis in various cancer types [30–32]. Low survivin expression was also correlated with statistically significant prolonged overall survival in a group of acute myeloid leukemia (AML) patients [33]. However, survivin expression in AML is heterogeneous and different transcription variants were reported to correlate with prognosis in various patient subsets. For example, the ΔEx3 variant is frequent in childhood leukemia, and its incidence correlates with shorter survival [34]. Contrarily, survivin variant 2B correlated with better survival in adult leukemia patients [34]. A response to chemotherapy depends on the presence of characteristic mutations. In particular, survivin promoter activity is potentiated by the AML1/ETO fusion gene [23,35], which characterizes AML with a t(8;21) subtype. Importantly, survivin downregulation mediated by MUC1-C/β-catenin pathway targeting was reported to sensitize AML cells to cytarabine (Ara-C) treatment [33]. Similarly, downregulation of survivin induced apoptosis in acute lymphocytic leukemia (ALL) cell lines and significantly potentiated the antileukemic effect of chemotherapy [36].

In this study, we report on enhanced survivin level in a panel of AML-derived cell lines treated with Ara-C, which is a part of standard AML treatment, called "7 + 3". In this regimen, Ara-C is combined with an anthracycline. From the most used anthracycline variants, doxorubicin was mostly found to attenuate survivin level [16,18,37], whereas idarubicin (IDA) treatment upregulated survivin [38,39]. Survivin silencing with small interfering RNA (siRNA) or short hairpin RNA (shRNA) potentiated an effect of chemotherapy on cell proliferation and survivin expression [35,36]. Therefore, targeting survivin with specific inhibitor may improve cancer treatment. The first and the most tested drug is a small-molecule survivin inhibitor YM-155 [40]. It inhibits survivin transcription through direct interaction with the promoter of survivin gene. Clinical trials investigating the safety and tolerability of YM-155 were conducted with encouraging results [41,42]. It was found to be effective in prostate carcinoma, melanoma, or NSCLC [43–45], albeit usually not as a single agent but in combination with chemotherapy. Several studies reported that YM-155 had more off-target effects, e.g., Mcl1 or XIAP inhibition [39,46], DNA damage [27], or NF-κB inhibition [47]. We tested an effect of YM-155 and its combination with Ara-C on the AML cell lines, and we found a complementary effect of IDA and YM-155 on cell viability and survivin expression. We conclude that IDA effectively reverses the effect of Ara-C on survivin expression in the majority of leukemia cell lines. Moreover, survivin suppression

occurs after YM-155 treatment in the cells resistant to IDA. Importantly, altered levels of other antiapoptotic proteins, Mcl1 and XIAP, considerably affect the response of AML cells to IDA or YM-155 treatment, either alone or in combination with Ara-C.

## 2. Materials and Methods

### 2.1. Cell Culture and Chemicals

Leukemia cell lines KASUMI-1, MV4-11, OCI-AML2, OCI-AML3, and KG-1 were purchased from DSMZ (Germany), while HL-60 cells were obtained from ECACC (GB). The cells were cultivated in growth media with fetal bovine serum (FBS), glutamine, and antibiotics (all from Sigma-Aldrich) according to the manufacturers' recommendations: MV4-11, KG-1, and HL-60 in RPMI-1640/10% FBS, OCI-AML2 and OCI-AML3 in alpha-MEM/20% FBS, and KASUMI-1 in RPMI-1640/20% FBS. All cells were cultivated from young stock in 5% $CO_2$ atmosphere at 37 °C, regularly checked for contamination. Stock solutions of 10 mM Ara-C, 5 mM IDA, and 5 mM YM-155 were added to cell suspensions to final concentrations and times as indicated in Section 3.

### 2.2. Cell Counting

Cell number and viability were assessed by cell counting in a Bürker chamber with the trypan blue uptake method. The cell suspension was mixed 1:1 with trypan blue solution (Sigma-Aldrich) and placed into a Bürker chamber. Cells in all large squares of the crosshatch were counted, and the viability was evaluated as a fraction of cells loaded with trypan blue.

### 2.3. Flow-Cytometry

Cell viability was measured using the propidium iodide (PI) exclusion test from aliquots of cell suspensions. A total of 200 microliters of cell suspension were mixed with 1 μL of 250 μM PI stock solution, incubated on ice for 5 min, and analyzed on a BD Fortessa flow cytometer. A fraction of PI-positive cells was evaluated to assess the cell viability.

### 2.4. Real-Time PCR

RNA from $5 \times 10^6$ cells was isolated using the RNeasy Mini Kit (Qiagen, Venlo, The Netherlands), and complementary DNA (cDNA) was generated by reverse transcription on CFX96 real-time system (BioRad) using a SensiFAST cDNA Synthesis Kit (Bioline, London, UK). Template RNA and resulting cDNA quality and concentration were assessed using the ND-1000 Nanodrop system (Thermo Fisher Scientific, Waltham, MA, USA). The relative amount of survivin was measured by real-time PCR (qPCR) using SensiFAST SYBR No-ROX Kit (Bioline) and calculated using Bio-Rad CFX Manager Software. The sequences of primer pairs designed to target survivin (transcript variant 1, NCBI RefSeq NM_001168.3) were as follows: forward CCACTGAGAACGAGCCAGAC, reverse TGTTCCTCTATGGGGTCGTCA. For relative quantification by the $2^{-\Delta\Delta Ct}$ method, GAPDH expression was measured as a reference, using GAAACTGTGGCGTGATGGC and CCGTTCAGCTCAGGGATGAC as the forward and reverse primers, respectively. The primer efficiency was checked according to the Pfaffl method, and it was found to be almost 100% for both survivin and GAPDH primer pairs.

### 2.5. Immunoblotting

Cells were washed with PBS and lysed in Laemmli sample buffer (SB, 50 mM Tris pH 6.8, 2% SDS, 100 mM DTT, 10% glycerol), boiled at 95 °C for 5 min, and centrifuged at $200,000 \times g/4$ °C for 4 h; the supernatant was stored at −20 °C. Five to ten microliters of each sample were subjected to SDS-PAGE and transferred to a PVDF membrane (BioRad). Mouse monoclonal antibodies against β-actin, survivin, caspase-3, XIAP, and p53, and rabbit polyclonals against PARP, Mcl-1, and p21 were from Santa Cruz Biotechnology. All primary antibodies were used at a dilution of 1:100–1:500. Anti-mouse and anti-rabbit HRP-conjugated secondary antibodies were purchased from Thermo Scientific and used

at concentrations 1:10,000–1:100,000. An ECL Plus Western Blotting Detection System (GE Healthcare, Chicago, IL, USA) was used for chemiluminescence visualization and evaluation by a G-box iChemi XT4 digital imaging device (Syngene Europe, Cambridge, UK).

### 2.6. Immunofluorescence

The samples were prepared as described previously [48]. Briefly, cells in suspension were seeded on a coverslip in humidified chamber for 15 min and then fixed with 4% paraformaldehyde (PFA) overnight at 4 °C. After 10 min of permeabilization by 0.5% Triton X-100, the cells were incubated for 1 h with a mouse anti-survivin primary antibody (Santa Cruz Biotechnology, Dallas, TX, USA, 1:100) and for another 1 h with the secondary antibody (Alexa-Fluor555-conjugated anti-mouse, Life Technologies, Carlsbad, CA, USA, 1:200) and with Hoechst33342 (1 μM, Life Technologies). The stained cells were observed under confocal laser scanning microscope FluoView FV1000 (Olympus Corporation, Shinjuku, Japan).

### 2.7. Statistical Analyses

Experiments were repeated at least three times and statistical evaluation was performed. A *p*-value of 0.05 or lower was preset to be indicative of a statistically significant difference between groups compared. In diagrams, arithmetic means of replicates of all experiments were plotted with SD error bars. Significance levels (*p*-values of two-way ANOVA) were determined using InStat Software (GraphPad Software).

## 3. Results

Survivin downregulation participates in apoptosis of myeloid leukemia cells induced by a combination of histone deacetylase inhibitor (SAHA) and DNA methyltransferase inhibitor (decitabine) [49]. Attenuated survivin expression was also found after all-*trans*-retinoic acid treatment (ATRA). In the present study, we focused on the survivin expression in leukemia cells treated with the drugs used in the first-line induction therapy protocol "7 + 3" for AML treatment, i.e., cytarabine (Ara-C) and idarubicin (IDA).

### 3.1. Ara-C Treatment

Expression of survivin was investigated in a panel of AML-derived cell lines (Figure 1a). We did not detect any substantial difference among the relative survivin expression levels in individual cell lines with respect to their characteristic gene mutations (AML1/ETO in Kasumi-1, Flt3-ITD in MV4-11, DNMT3A in OCI-AML2 and OCI-AML3, NPM1 in OCI-AML3, or p53 in Kasumi-1, KG-1 and HL-60). In accordance with our previous results [49], ATRA induced attenuation or no change in survivin expression in the cell lines (Figure 1b and Figure S2). On the other hand, Ara-C caused survivin overexpression in a broad interval of concentrations (50 nM–5 μM, Figure 1b). It also dramatically slowed the proliferation (Figure 1c), but a large drop in viability was observed only in HL-60 and OCI-AML2 at higher concentrations (Figure 1d).

Expression of survivin is cell-cycle-regulated and reaches its maximum in mitosis [16]. Ara-C is known to arrest cells in S phase [50,51]. Enhanced survivin expression, thus, cannot be attributed to altered fraction of mitotic cells. Indeed, we detected increased survivin level in the nuclei of interphase cells and, simultaneously, we did not find any mitotic cells in Ara-C-treated samples (Figure 2).

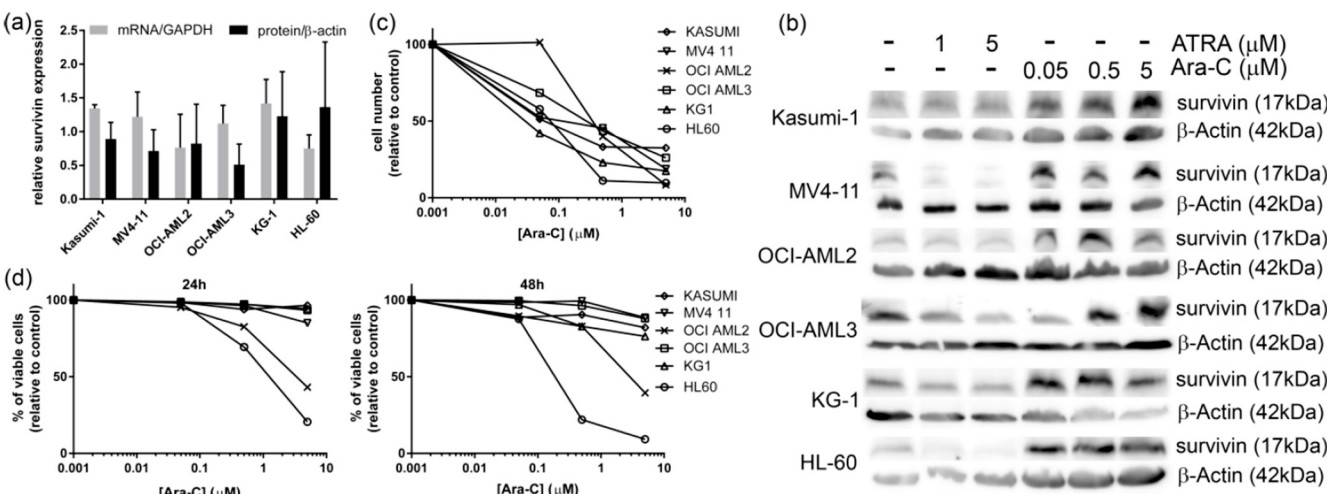

**Figure 1.** (**a**) Relative survivin expression at messenger RNA (mRNA, gray) and protein (black) level in a panel of acute myeloid leukemia (AML) cell lines. Values represent an average of at least three independent measurements ± SD (**b**) Effect of 24 h all-*trans*-retinoic acid treatment (ATRA) or cytarabine (Ara-C) treatment on survivin expression. β-Actin levels served as the loading control. (**c**) Cell proliferation after 48 h of Ara-C treatment. (**d**) Cell viability assessed by trypan blue uptake after 24 h (left) or 48 h (right) Ara-C treatment. Original blots with molecular weight (MW) markers are provided in the Supplementary Materials, Figures S1 and S2

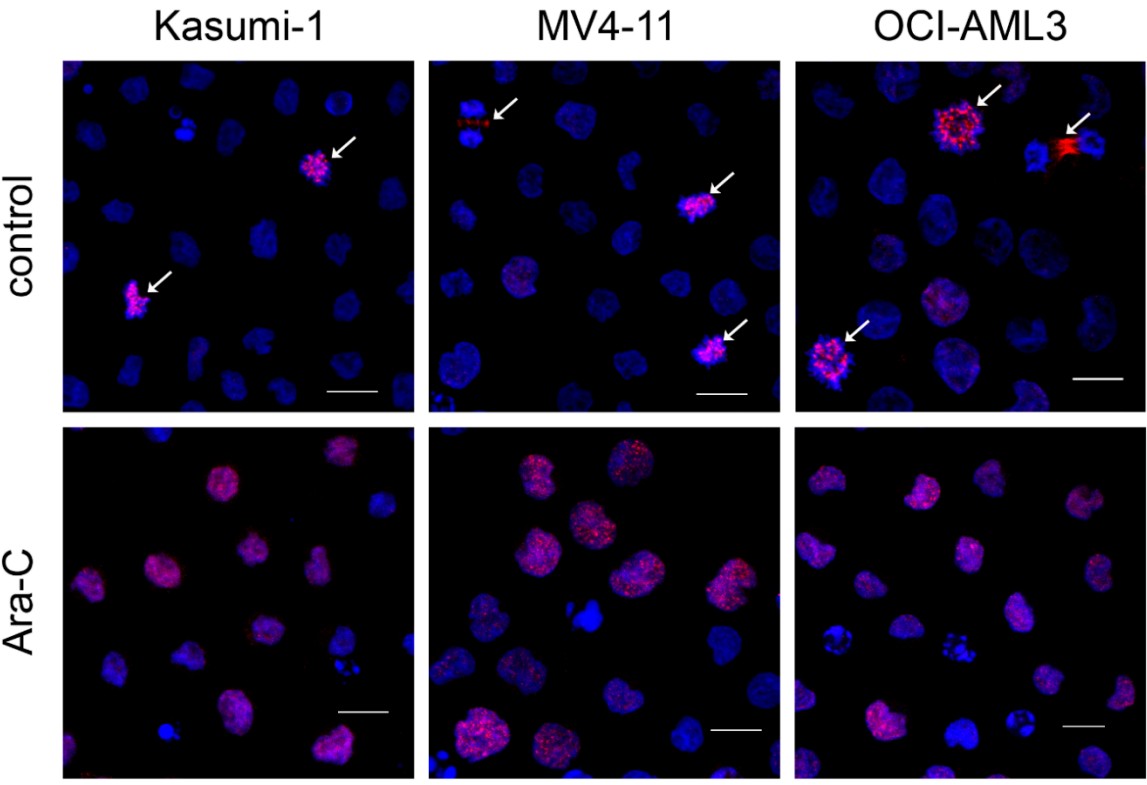

**Figure 2.** Immunofluorescence detection of survivin expression (Alexa Fluor555, red) after 24 h in control (upper row) and 5 µM Ara-C-treated (lower row) AML cell lines. Nuclei are counterstained with Hoechst33342. Arrows indicate mitotic cells. Bars represent 10 µm.

In subsequent experiments, we used 5 µM Ara-C for all cell lines except for HL-60, which was considerably more sensitive; thus, it was treated with 0.5 µM Ara-C.

### 3.2. Combination with IDA

The second component of the standard "7 + 3" AML therapy regimen is an anthracycline drug (most frequently, daunorubicin or idarubicin). From a dose–response viability assay (see Supplementary Materials, Figure S1), a 100 nM concentration of IDA was chosen for further experiments. The effect of anthracycline drugs on survivin expression is poorly investigated, and studies reporting on both survivin level enhancement and attenuation can be found [18,39]. Accordingly, we observed both these effects of 100 nM IDA treatment on survivin expression depending on the cell line (Figure 3a,b and Figure S2). Ten times lower IDA concentration was used for HL-60 cell line treatment as this line was reported to be very sensitive to IDA treatment [52]. Interestingly, cell viability was reduced by IDA treatment in all cell lines independently of survivin expression (Figure 3a) and PARP fragmentation was simultaneously detected (Figure 3b). We, therefore, tested a set of other apoptosis-related proteins, and we revealed considerably reduced expression of another member of the IAP family, XIAP (Figure 3b). Moreover, Mcl1 was suppressed in all cell lines except OCI-AML3 and expression of p21 increased in the cell lines with functional p53 (Figure 3b). Interestingly, upregulation of p53 was detected in MV4-11 and OCI-AML3 but not in OCI-AML2.

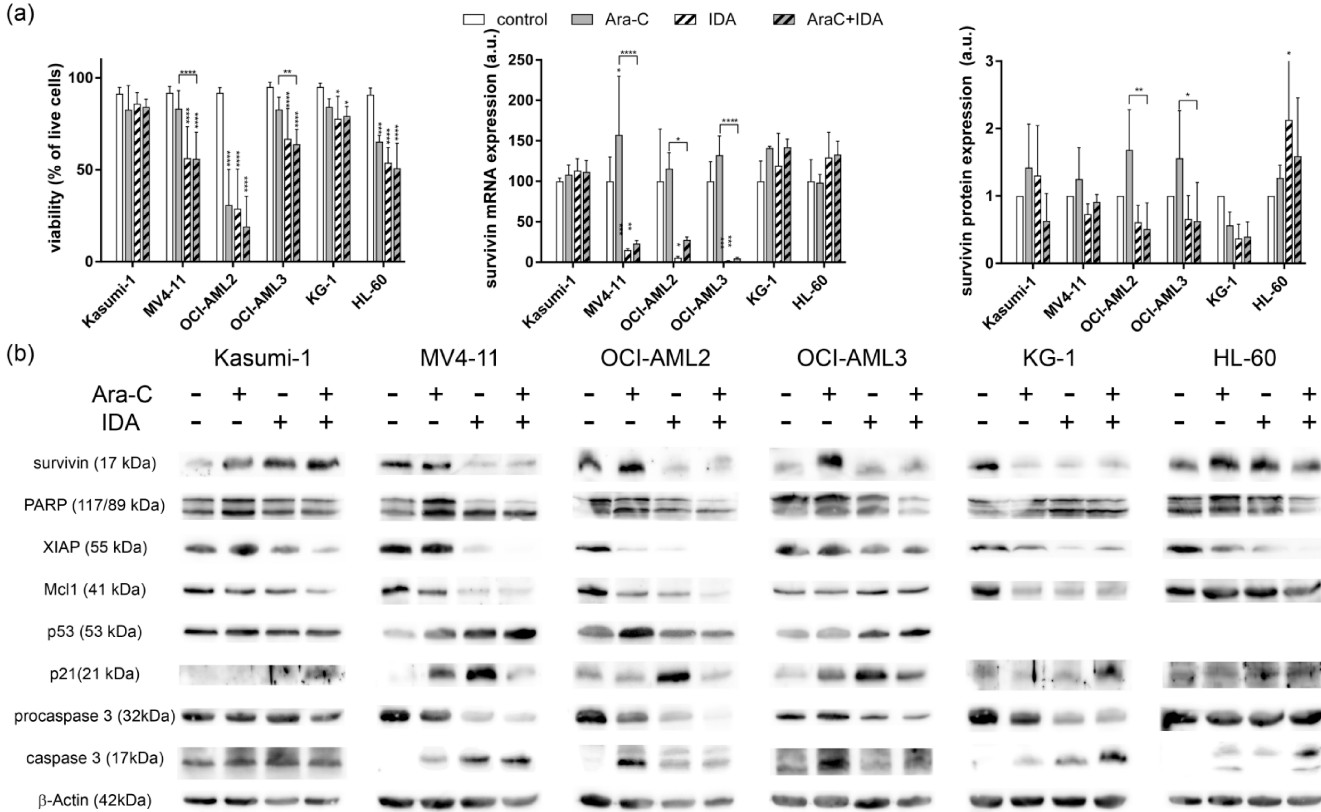

**Figure 3.** Effect of 100 nM (10 nM for HL-60) idarubicin (IDA) and its combination with 5 μM (0.5 μM for HL-60) Ara-C on AML cell lines. (**a**) Viability (estimated by propidium iodide (PI) exclusion test) and survivin mRNA and protein expression in control (white), Ara-C (gray), IDA (white, hatched), or Ara-C + IDA (gray, hatched)-treated cells. Values represent an average of at least three independent measurements ± SD. Significance of the difference is marked by asterisks (difference vs. control is indicated above the appropriate bar): * $p < 0.05$, ** $p < 0.01$, *** $p < 0.001$, **** $p < 0.0001$. (**b**) Expression of apoptosis-related proteins. Blots are representatives of at least three independent experiments. Original blots with MW markers are provided in the Supplementary Materials, Figure S2.

Combination of Ara-C and IDA did not show any synergism of these two drugs in terms of the effect on cell viability. However, Ara-C-induced survivin overexpression was reversed by the combination with IDA in the lines where single IDA treatment

caused survivin depletion. Similar results were obtained also at the survivin mRNA level (Figure 3a). Immunofluorescence monitoring of survivin expression confirmed the results of immunoblot (Figure 4 and Figure S3) and revealed no mitotic cells in IDA-treated samples, although IDA induces G2/M (mitosis) cell-cycle arrest after 24 h [52].

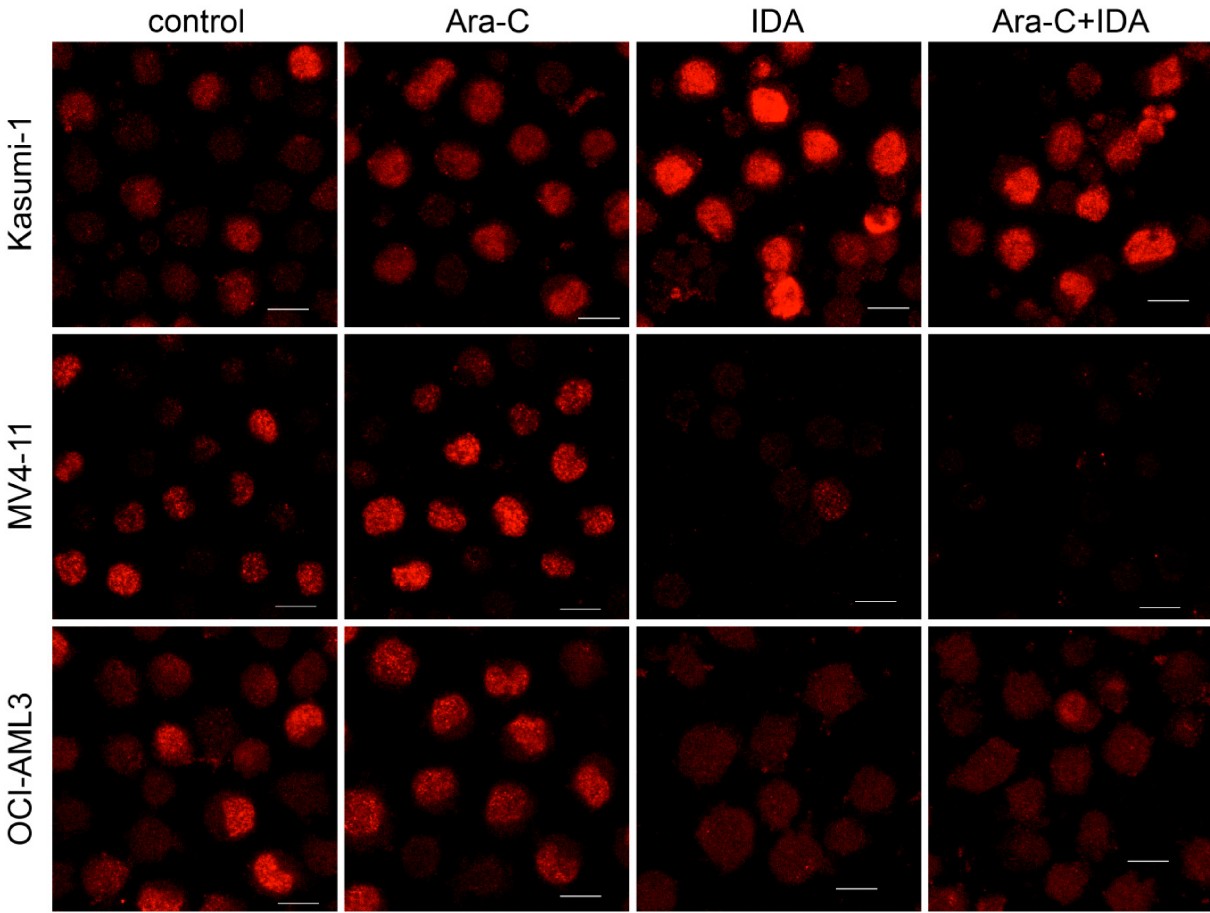

**Figure 4.** Immunofluorescence detection of survivin expression (Alexa Fluor555, red) in control, 5 µM Ara-C and/or 100 nM IDA-treated AML cell lines. Bars represent 10 µm.

### 3.3. Survivin Inhibition with YM-155

As the effect of IDA on the panel of AML cell lines was heterogeneous, we further targeted the survivin expression with 200 nM YM-155, a small molecule which binds to the survivin promoter and regulates its transcription [40]. Again, as the HL-60 cell line was reported to be extremely sensitive to YM-155 [46,53], we used 10 times lower concentration for this line. Interestingly, YM-155 attenuated both mRNA and protein survivin expression in the majority of AML cell lines except MV4-11 (Figure 5a,b and Figure S2). Similarly, as with IDA, a viability drop was detected in all YM-155-treated cells despite the variability in survivin expression changes (Figure 5a). Combination with Ara-C did not significantly alter these effects. We also investigated the expression of the other proteins and we found marked downregulation of XIAP in the majority of cells except OCI-AML3. p53 expression was mostly not affected by the YM-155 treatment and Mcl1 and p21 were differently regulated in individual cell lines (Figure 5b).

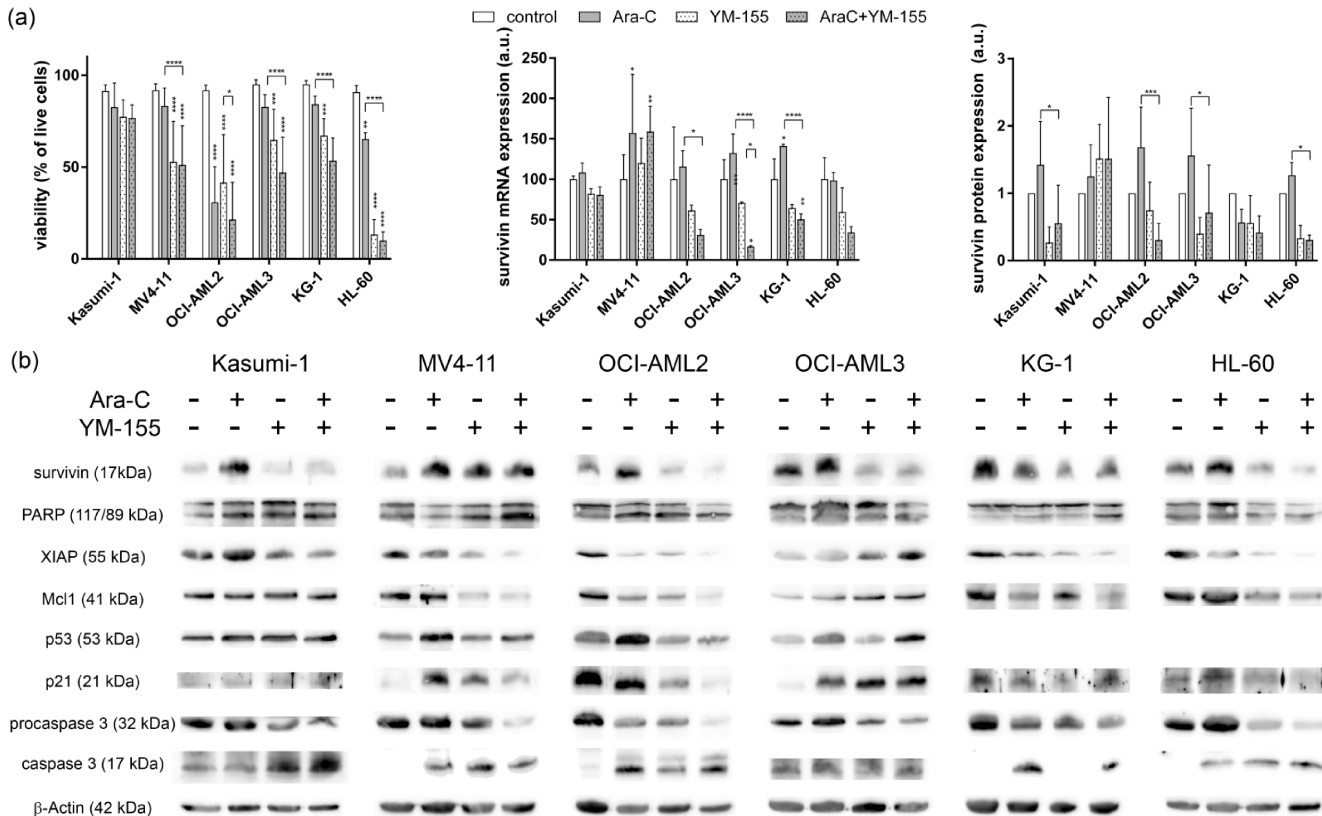

**Figure 5.** Effect of 200 nM (20 nM for HL-60) YM-155 and its combination with 5 μM (0.5 μM for HL-60) Ara-C on AML cell lines after 24 h treatment. (**a**) Viability (estimated by PI exclusion test) and survivin mRNA and protein expression in control (white), Ara-C (gray), YM-155 (white, dotted), or Ara-C + YM-155 (gray, dotted)-treated cells. Values represent an average of at least three independent measurements ± SD. Significance of the difference is marked by asterisks (difference vs. control is indicated above the appropriate bar): * $p < 0.05$, ** $p < 0.01$, *** $p < 0.001$, **** $p < 0.0001$. (**b**) Expression of apoptosis-related proteins. Blots are representatives of at least three independent experiments. Original blots with MW markers are provided in the Supplementary Materials, Figure S2.

Decreased survivin expression in the nuclei of interphase cells was confirmed by immunofluorescence (Figure 6 and Figure S3).

Contrary to Ara-C or IDA treatment, YM-155 did not affect the ability of cells to enter mitosis. As we reported earlier [49], a combination of epigenetic drugs 5-aza-2′-deoxycytidine (decitabine) and suberoylanilide hydroxamic acid (SAHA) affected the centromere-related position of survivin in the mitotic cells; however, the localization of survivin was not altered by YM-155 treatment suggesting that YM-155 has no impact on the cell-cycle-related survivin function (Figure 7). Nonetheless, addition of Ara-C to YM-155 resulted in loss of the mitotic fraction in all cell lines.

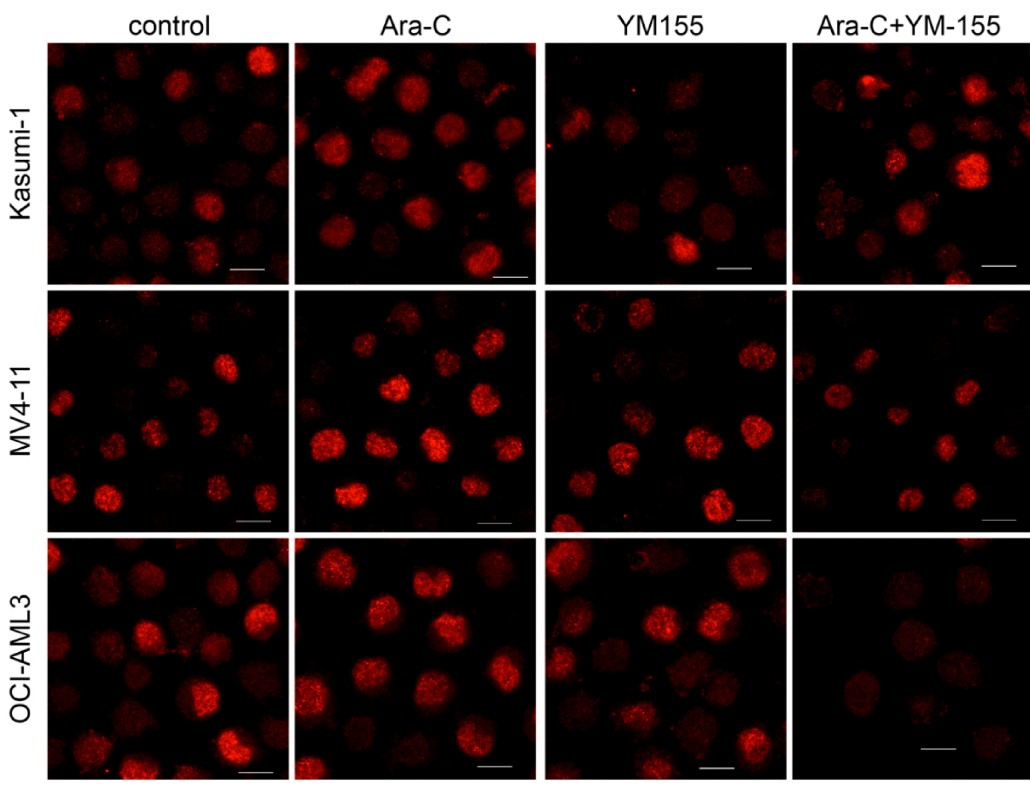

**Figure 6.** Immunofluorescence detection of survivin expression (Alexa Fluor555, red) in control, 5 µM Ara-C and/or 200 nM YM-155-treated AML cell lines. Bars represent 10 µm.

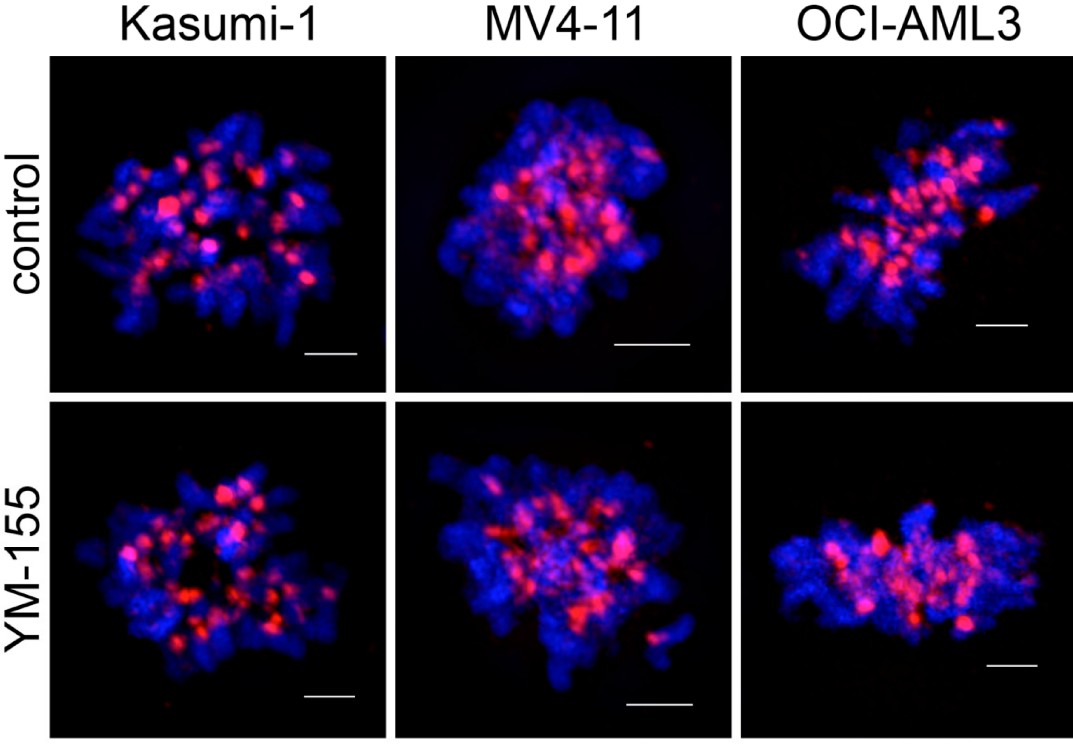

**Figure 7.** Effect of YM-155 on survivin (AlexaFluor555, red) localization in mitotic cells. Chromosomes are stained with Hoechst 33342 (blue). Bars represent 2 µm.

## 4. Discussion

Survivin belongs to the family of inhibitors of apoptosis and contributes to cell survival by blocking the activation of executive caspases [8]. However, survivin signalization is rather complex, as it simultaneously ensures proper cell division [7], and the variability of these roles is reflected by versatile survivin activity. We showed, in this study, that survivin is upregulated in the nucleoplasm of AML cells treated with Ara-C (Figure 2), although Ara-C considerably lowers cell proliferation and slightly decreases cell viability (Figure 1). The mechanism of Ara-C-induced survivin upregulation remains unknown. However, it was reported in hematopoietic cells that low dose of Ara-C induced S-phase arrest followed by an activation of cell cycling after 72 h [50]. Simultaneously, survivin downregulation sensitized AML cells to cytarabine treatment [33]. We, thus, speculate that increased survivin expression contributes to overcome a transient lack of cytidine trisphosphate during a single dose of Ara-C treatment. A combination of Ara-C with an anthracycline is used to treat most of AML cases. We, thus, measured an effect of IDA on survivin level in a panel of AML cell lines and we detected different survivin regulation in individual cell lines (Figures 3 and 4, discussed below in detail). We also investigated an impact of specific survivin inhibitor YM-155 on the same set of AML lines, and we found attenuated survivin levels in the lines which did not downregulate survivin after IDA treatment (Figures 5–7). Simultaneously, we observed altered expression of several other apoptosis-related proteins, particularly XIAP, Mcl1, p53, and p21. No correlation was found between the protein content of cultivation medium (% of FBS), recommended for individual cell lines by the providers, and an effect of cytostatics.

Survivin expression was found to be activated by the AML1/ETO fusion gene [35], which is present in the Kasumi-1 cell line. Simultaneously, enhanced survivin and XIAP levels were suggested to account for the IDA resistance of the Kasumi-1 cell line [39]. On the other hand, survivin downregulation by YM-155 was observed in this cell line [22,53]. Our results confirmed the IDA-induced survivin overexpression and drug resistance, as well as survivin level attenuation, after YM-155 treatment in the Kasumi-1 cell line. A moderate decrease in XIAP expression was observed in both cases. On the other hand, we detected no change in Mcl1 expression and a stable high expression of p53, which is mutated (R248Q) in Kasumi-1 cells. Pterocarpanquinone LQB-118, an inhibitor of IAP proteins, induced deregulation of survivin and XIAP in Kasumi-1, as well as in primary cells [39]. However, whereas XIAP attenuation was observed together with apoptosis, inhibition of survivin expression alone did not correlate with apoptosis induction [39]. In summary, complex IAP inhibition, but not that of the survivin alone, seems to be a promising alternative treatment for these IDA-resistant cells.

Survivin expression is regulated by the activity of STAT3/5 and β-catenin, which are both targets of Flt3. Internal tandem duplication in Flt3 (Flt3-ITD), one of the AML-characteristic mutations, constitutively activates Flt3 kinase, resulting in aberrantly high phosphorylation of STAT3/5 and survivin overexpression [24]. Targeting Flt3 activity was shown to decrease survivin level [24,54]. This suggests that survivin inhibition may bring some benefits to AML patients with Flt3-ITD. MV4-11, a cell line derived from an AML patient with Flt3-ITD, exhibited relatively high survivin expression. Despite a substantial effect on cell viability, YM-155 did not affect survivin expression in this line (Figure 5), indicating superiority of the Flt3-ITD-induced survivin overexpression over the YM-155-mediated inhibition. However, YM-155 considerably lowered XIAP and Mcl1 expressions, likely owing to a known off-target effect [39,46]. Interestingly, survivin, together with XIAP and Mcl1, was attenuated in IDA-treated MV4-11. In addition, IDA-induced apoptosis was accompanied by increased levels of tumor suppressor p53 and its target protein p21 in MV4-11 cells.

A large drop in viability and PARP fragmentation were induced by all drugs in the OCI-AML2 cell line derived from an AML patient with DNMT3A R635W mutation. Ara-C-induced p53 expression was not followed by an increase in p21, but the simultaneous drop in XIAP and Mcl1 expressions likely contributed to extensive apoptosis despite survivin

upregulation. Conversely, p53-independent p21 upregulation occurred after IDA treatment together with survivin, XIAP, and Mcl1 decrease. A similarly unusual p53 regulation in the OCI-AML2 cell line, which has no reported p53 mutations, was also previously observed after NSC348883 treatment [55]. Combination of Ara-C with IDA canceled the p21 overexpression and further enhanced a fraction of dead cells. YM-155 substantially decreased the survivin, XIAP, and Mcl1 levels and, in combination with Ara-C, it also caused p21 deregulation, leading to extensive apoptosis.

In the OCI-AML3 cell line (with NPM1 mutation type A and DNMT3A R882C), survivin deregulation correlates with a viability drop induced by IDA or YM-155. Moreover, Ara-C and IDA upregulate p53, and extensive p21 induction is observed in all samples. Simultaneously, XIAP and Mcl1 remain unchanged, suggesting that survivin regulation plays an important role in apoptosis in these cells. Whether it may have a consequence with aberrant localization of mutated NPM remains to be investigated.

Survivin mRNA upregulation after IDA was observed in two p53-null cell lines, KG-1 and HL-60. Whereas, in HL-60, the protein level also increased, KG-1 cells showed a decrease in survivin after IDA treatment. Different regulation of survivin at the mRNA vs. protein level has also been documented in esophageal cancer cells [12]. Interestingly, the viability was only slightly decreased in KG-1 although XIAP and Mcl1 expressions were substantially lowered. On the contrary, a high fraction of dead cells occurred in IDA-treated HL-60 cells although no considerable changes were detected in XIAP, Mcl1, and p21 levels. Importantly, YM-155 caused a viability drop, together with PARP fragmentation and survivin, XIAP, and Mcl1 downregulation in both these p53-null cell lines, and its effect was potentiated by combination with Ara-C. YM-155, thus, reliably reverts Ara-C-induced survivin upregulation and triggers apoptosis in p53-null cell lines.

In accordance with the results shown in the literature [53], the level of antiapoptotic protein Bcl2 was unchanged (data not shown). Moreover, we did not notice any significant change in expression of NF-κB, which was reported to participate in response to YM-155 treatment in several tumor types [8,16].

Lastly, immunofluorescence (IF) staining confirmed that enhanced survivin expression detected by immunoblot reflects a higher occurrence of survivin in the nuclei of interphase cells. We did not detect any mitotic cells in Ara-C- and/or IDA-treated cells. On the other hand, mitosis was not affected by YM-155 treatment, and localization of survivin in mitotic cells was identical in control and YM-155 samples. Therefore, YM-155 probably does not compromise the proliferation of normal cells, which corresponds with its good tolerability. Interestingly, GFP-tagged survivin is often found in the cytoplasm of transfected cells [56,57], whereas we found prevalently nuclear localization in the interphase cells by immunofluorescence staining of endogenous survivin. As our finding correlates with IF data provided by various survivin antibody suppliers (Abcam, Cell Signaling Technology, Santa Cruz BioTechnology et al.), we suggest that, if the GFP tag is used for small proteins, attention must be paid to pitfalls eventually arising from aberrant localization of the resulting fusion protein.

In summary, we showed that, although Ara-C, IDA, and YM-155 lower cell proliferation and induce apoptosis in almost all cases, mechanisms leading to apoptosis clearly differ among individual AML cell lines, likely reflecting the heterogeneity of the disease. Our results are in accordance with the previously published study reporting on heterogeneous proliferation and viability changes induced by survivin-targeting siRNA in several AML cell lines [53]. A heterogeneous response on YM-155 treatment was also noted in ALL [47]. The authors of that study suggested substituting doxorubicin with YM-155 in preselected ALL patients in order to eliminate the adverse side effects of doxorubicin treatment. A study combining selective survivin inhibitor LY2181308 with Ara-C + IDA treatment also showed some benefits, particularly for AML patients with high initial survivin level [58].

## 5. Conclusions

Treatment with Ara-C induces survivin expression in a panel of AML cell lines. Simultaneous arrest in the S phase of the cell cycle suggests that ara-C-induced survivin overexpression should be attributed to its antiapoptotic function. Idarubicin, the second component of the gold standard for AML therapy, causes survivin downregulation in some, but not all, cell lines. Treatment of IDA-resistant cells with the survivin inhibitor YM-155 results in attenuated survivin expression in most cell lines. Apoptosis caused by the IDA or YM-155 is accompanied with changes in the levels of antiapoptotic proteins XIAP and Mcl1, the tumor suppressor p53, and cyclin-dependent kinase inhibitor p21. In contrast to Ara-C-induced survivin overexpression, changes induced by IDA or YM-155 considerably vary among individual AML cell lines and likely mirror the substantial heterogeneity of the disease. However, targeted survivin inhibition offers a possibility to overcome the resistance to standard AML treatment protocol in selected patients, which did not respond to idarubicin therapy. Moreover, substitution of IDA with a specific survivin inhibitor would bring a benefit in better tolerability for a specific subset of AML patients.

**Supplementary Materials:** The following are available online at https://www.mdpi.com/2076-3417/11/1/460/s1: original blots for the figures containing Western blot results.

**Author Contributions:** B.B. designed the study, performed Western blotting, confocal microscopy, flow cytometry, and statistics, and wrote the manuscript; P.O. performed PCR experiments and analyses. All authors have read and agreed to the published version of the manuscript.

**Funding:** The work was supported by the Czech Science Foundation (grant No 19-04099S) and the Ministry of Health of the Czech Republic (project for conceptual development of the research organization No 00023736).

**Institutional Review Board Statement:** Not applicable.

**Informed Consent Statement:** Not applicable.

**Data Availability Statement:** The data presented in this study are available in Supplementary Materials, see above.

**Conflicts of Interest:** The authors declare no conflict of interest.

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
