# Peer review of "Chemotherapy-Induced Survivin Regulation in Acute Myeloid Leukemia Cells"

_applsci, doi:10.3390/app11010460_

Round 1

Reviewer 1 Report

In this manuscript Otevrelova et al. describe the effect of standard chemotherapy cytarabine and idarubicin on the expression of survivin.   This is an interesting topic and any sound improvement to the current chemotherapy regimen and understanding of chemotherapy resistance would be greatly appreciated in the field. 

I would advise the authors to explain in more detail the rationale of their experiments, to add some technical details and revise their manuscript for clarity. The authors might want to provide further methodological details and molecular insights and resubmit the manuscript.

  • In Figure 1A the authors show the relative expression of Survivin in several AML cell lines. Are the data expressed as expression of Survivin/housekeeping gene? In the material and methods it is reported that the qPCR data were analysed by 2Delta Delta CT method. This is acceptable if the primer efficiency of target and housekeeping gene is comparable and almost 100%. Did the authors checked the efficiency of their primers? If the efficiency is not comparable the authors should use the Pfaffl equation.
  • In Figure 1 the authors report the effect of Cytarabin on survival of leukemic cells. I understand that in Figure 1B the authors report the cell number upon incubation for 48hrs, whereas in the lower panel they report the percentage of cells. I believe these are the same data, simply shown in two different ways. The authors might decide to show the data only in one way.
  • In Figure 2 the authors report the expression of Survivin by immunofluorescence and indicate mitotic cells. How can the authors identify mitotic cells without a specific marker? I would suggest to use one such as phosphoH3 to investigate whether Survivin is expressed in mitotic cells.
  • Figure 3 and 5 lack of statistical analysis, as well as technical details (drug concentration and incubation time).
  • Figure 3 and 5 assess the effect of drugs on survival. The assay might not be fit to detect potential synergism between the two drugs. Detection of apoptotic cells, by Annexin V staining can be used to investigate this aspect.
  • In figure 3 and 5 It is not clear why the authors assessed the expression of p53, p21 and MCL1 in response to drug treatment rather than a specific apoptotic marker, such as the activation of caspase 3/8, which is also dependent on Survivin expression.
  • Figure 7, can the authors add some technical information? Is the expression of Survivin nuclear? How can the authors identify mitotic cells?
  • The overall conclusion does not fit with the results. Survivin expression seems to be heterogenous in the cell lines and it does not correlate with response to drug treatment. In addition, in this context, there is no enough evidence that the expression of Survivin is correlated to its anti-apoptotic function.

Reviewer 2 Report

In this study the authors provided a comparative analysis on the expression levels of survivin in a panel of AML cell lines treated with Ara-C, IDA and YM-55. They showed that the treatment with Ara-C induces an increase of expression levels of survivin, and caused the growth arrest and depletion of mitotic cells fraction. IDA reversed Ara-C – induced survivin upregulation in  majority of AML cell lines, whereas YM-155 caused survivin deregulation together with a viability  decrease in cells resistant to IDA. 

This study is well described and clearly presented. Moreover, results obtained are well discussed and conclusions are supported by the results.

I believe that the paper can be published in the present form.

Reviewer 3 Report

The manuscript by Brodska et al attempts to demonstrate a critical role for survivin in AML response to chemotherapy treatments.  Identifying new therapeutic combinations for AML treatment and understanding why AML cells become resistant to standard of care therapeutics is a very important endeavor.  However, the work presented herein are not likely to have an impact and the data have several issues that would preclude it from publication.  Below is a list of some of the issues detected with the work:

major concerns:

  1. Western blots are of of high quality, are over-cropped and do not have any molecular weight markers.
  2. There are absolutely no statistical analyses performed anywhere in the manuscript
  3. the IF figures (2, 4, 6, 7) are not informative.  the quality of the images is poor, the appropriate controls are not presented, no counter stains are used, there is no quantification of data.
  4. Most of the data presented are so variable between cell lines that making conclusions is nearly impossible.
  5. All data provided is phenominology and as presented is only qualitative observations.

Minor concerns:

1.  a dose response with IDA should be provided.

2.  dose daunorubicin or doxorubicin behave similarly to IDA in these analyses?

3.  it is unclear why graphs are not normalized to controls?  for example in figure 1C, how can the '0' treatments start at 50% on the y-axis for some cell lines? also, graphs showing expression (mRNA and protein) would be more easily interpreted if they were normalized to the untreated condition for each individual cell line.

Round 2

Reviewer 3 Report

the manuscript is improved from last submission.  However, I personally still diagree with the over-cropped blots and the lack of actual molecular weight markers.  figure 1 still has no markers and the other figures simply put the predicted size of the protein in parentheses, which is not appropriate.  i am still not sure what information is gained from the data presented in figure 7 
